# Vision-Based Design and Deployment Criteria for Power Line Bird Diverters

Graham R. Martin

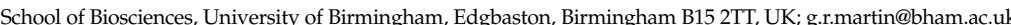

School of Biosciences, University of Birmingham, Edgbaston, Birmingham B15 2TT, UK; g.r.martin@bham.ac.uk

**Simple Summary:** Aspects of the vision of birds pertinent to the design and deployment of power line bird diverters are described. Their application is illustrated using Canada Geese as a putative worst-case example of a species prone to power line collision.

**Abstract:** The design of bird diverters should be based upon the perception of birds, not the perception of humans, but until now it is human vision that has guided diverter design. Aspects of bird vision pertinent to diverter design are reviewed. These are applied in an example that uses Canada Geese *Branta canadensis* as a putative worst-case example of a collision-prone species. The proposed design uses an achromatic checkerboard pattern of high contrast whose elements match the low spatial resolution of these birds when they are active under twilight light levels. The detectability of the device will be increased by movement, and this is best achieved with a device that rotates on its own axis driven by the wind. The recommended spacing of diverters along a power line is based upon the maximum width of the bird's binocular field and the linear distance that it subtends at a distance sufficient to allow a bird to alter its flight path before possible impact. Given the worst-case nature of this example, other bird species should detect and avoid such a device. The basic design can be modified for use with specific target species if sufficient is known about their vision. Field trials of devices based on these design criteria are now required.

**Keywords:** birds; vision; acuity; binocular vision; Canada Goose; collision mitigation

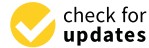



## 1. Introduction

It would seem obvious that the design of bird diverters should be based on the perception of birds. However, the designs of bird diverters have been based on the perception of humans and on engineering considerations, such as ease of deployment and weight. There appears to have been no attempt to employ a bird vision perspective in the design of power line bird diverters or in recommendations for their spacing along a power line.

Until recently, however, a bird's eye view of the world was perhaps not readily understood and there was insufficient knowledge about bird vision to guide diverter design. It is argued here, however, that there is now sufficient knowledge about bird vision [1,2] to tease out some basic parameters that can be used to guide the design and deployment of diverters. Detailed information is available on the vision of only a relative handful of the 11,000 extant bird species. Therefore, generalizations have to be made with caution about the general properties of bird vision and their application to diverter design and deployment. However, there now seems to be sufficient information on some key parameters of bird vision to guide diverter design and deployment. The aim here is to suggest design and deployment guidelines that can have general applicability across species and locations.

## 2. Factors to Be Taken into Consideration in the Design of Bird Diverters

It is not sufficient that a diverter "looks diverting" to the human eye. A diverter must be detectable by a bird, or more properly a range of bird species, under a range of

light and visibility conditions. Furthermore, when a bird detects the diverter its behaviour should change, the primary aim being to redirect the bird's flight path sufficiently to avoid a collision with the diverter, and with the power line to which it is attached.

Humans tend to fall readily into the trap of believing that the world as they perceive it, is how the world is. Humans, however, like all species, are prisoners of their senses which have been shaped by natural selection to gain information for the conduct of a range of tasks that are specific to their natural ecology. While human perception of the world provides a persuasive unified view, it is only one worldview. Each animal species extracts different information from the external world. Even closely related species, including birds within the same family, will have different information available to them related to their particular behaviour and ecology [1,2].

Discussed below are some general aspects of the vision of birds that should be considered in the design and deployment of bird diverters. Appropriately designed and deployed bird diverters, which result in reduced collision occurrences, should be welcomed from the perspective of the welfare of birds and by power utilities. Factors discussed here should help broaden the basis for power line diverter designs and should provide general solutions for diverter design aimed at a range of collision-prone bird species.

### 2.1. Vision and Visual Capacities

Vision is the prime sense used by animals to retrieve information about the world that surrounds them. At its most fundamental, vision has evolved to provide information about where objects are over a range of distances [3]. This is true for humans as much as it is for all other species, but the information that can be extracted about objects, and where they are, varies significantly between species. Furthermore, in every species, the accuracy of spatial information also varies significantly with light level, and hence time of day. Animals, including humans, are hostage to the sun and the moon as regards the information that they have available to them about the world

Vision is a complex and multifaceted sense, making it impossible to give a simple summary of the visual ability of any species. To describe vision, it is necessary to break it down into various "visual capacities" which need to be investigated and described separately. The most notable of these visual capacities are spatial resolution, colour vision, speed of vision, and field of view. The first three of these are highly dependent upon the light level and upon position within the field of view. Under natural conditions, light levels change dramatically within the daily cycle (typically 3000-fold between noon and twilight, and up to 100,000-fold at night depending upon the moon's presence and phase [1]. This means that the ability to retrieve information about objects from a scene varies throughout every daily cycle to a significant degree.

### 2.2. Changing Flight Behaviour

Like vision, behaviour is complex and multifaceted, there are multiple ways of describing and measuring behaviour. Power line diverters seek to change a quite specific part of a bird's repertoire at a particular location. While diverters seek to alter the flight path of birds it is necessary to recognize that flight, like vision, is multifaceted and differs between species. Each bird species has a specific flight pattern, manoeuvrability, and characteristic speed of powered flight [4,5]. Bird species also differ in the time of day at which they are most active or undertake flights in their daily cycle. One species will fly in twilight, while another will not fly until the sun is well above the horizon, and some species will fly during nighttime. Some have flight which is slow and highly manoeuvrable, while others fly fast and cannot readily change their trajectories [5].

### 2.3. Will a Diverter Divert?

The application of knowledge of vision and flight behaviour to the design of bird diverters can be reduced to one key question: will the diverting device be detected at a distance that should allow a timely change in flight path? Key to answering this question is

an understanding of which species a diverting device is aimed at. Additionally, at which locations, at what time of day, and under what weather conditions should the device have its desired effect? Weather conditions and light levels become important because they will determine what can be detected and at what distance.

In this context, it is worth reflecting upon our own perception of objects. What we see/detect, and how we interpret a natural scene, depends crucially upon the light level and weather conditions. A telling example of how these conditions are relevant to our behaviour, indeed our well-being, is car driving. What we detect, and how we interpret information when driving, has been subject to much research [2]. When driving we are constantly seeking new information about objects at a distance from us, towards which we are travelling. Whether we detect an object, and whether we interpret it correctly, depends significantly upon light levels and weather conditions [6,7].

Safe driving depends upon constant updates of information about the road ahead and interpreting changes correctly. Particularly important is the fact that we often, perhaps typically, rely upon partial information and frequently drive beyond our perceptual limit [7]. That is, we often do not have full information to determine what lies ahead, but we rely upon the predictability of roads and their standard design features to allow us to make judgements based on partial information.

Usually, we manage to behave correctly even though we have partial information from our vision. However, if conditions become exacting, and the acquisition of information is impaired, accidents can occur, even partial information may not be available and correct interpretations are not possible. In these more exacting circumstances, we need repeated information to warn us that the world ahead is changing before we fully recognize the true nature of the situation and change behaviour accordingly.

There would seem to be parallels here with the situations posed to birds by power lines and other obstacles that intrude into the air space in which they fly. Under these circumstances, birds are also likely to be relying upon partial or incomplete spatial information, as some species are known to do when foraging [1,8]. Natural objects that intrude into a bird's open flight space are rare and it is reasonable that the bird will often predict that there is little likelihood of an obstacle ahead. Thus, warning of a power line ahead may require significant and apparently excessive signals, much as is required to warn us that an obstacle lies ahead when driving on an open or fast road. Furthermore, when environmental conditions reduce the information available to the birds (particularly when light levels are low, and visibility is reduced) it may require significant and repeated signals to warn the birds of the obstacle and change their behaviour.

### 2.4. Visual Spatial Resolution and Light Level

In all animals, visual-spatial resolution decreases significantly with light level [3]. The examples of visual acuity in a range of bird species given in Table 1 apply to high daytime light levels, approximately those that occur in open habitats around midday under the full sun. Changes in spatial resolution within the range of light levels experienced during the day are typically small (heavy cloud reduces light levels by about 10-fold), however, as light levels fall into twilight a significant decrease in resolution occurs (Figure 1). At sunset, light levels have fallen by about 300-fold compared with noon and fall by another 100-fold during the twilight period. These changes in light level have a significant impact on spatial resolution (Figure 1).

**Table 1.** Spatial resolution (visual acuity) in a sample of bird species and in humans. All measures refer to acuity at high daytime light levels and indicate the best performance (finest spatial detail that can be resolved using high contrast stimuli). To compare between species simply divide their acuity values. Thus, the goose's resolution is about 7 times higher than that of a young human. Expressed another way, a young human should be able to detect a given object 7 times further away than a goose, while an older human can detect the same object 3 times further away.

| Species | Visual Acuity (Minutes of arc) | Source |
|---|---|---|
| Wedge-tailed Eagle *Aquila audax* | 0.2 | [9] |
| Brown Falcon *Falco berigora* | 0.4 | [10] |
| Rock Dove *Columba livia* | 1.7 | [11] |
| Budgerigar *Melopsittacus undulatus* | 2.6 | [12] |
| Canada Goose *Branta canadensis* | 3.1 | [13] |
| Bourke's Parrot *Neopsephotus bourkii* | 3.2 | [12] |
| Great Horned Owl *Bubo virginianus* | 4 | [14] |
| House Sparrow *Passer domesticus* | 6.3 | [15] |
| Human (young) | 0.4 | [3] |
| Human (mature/with spectacles) | 1.0 | [3] |

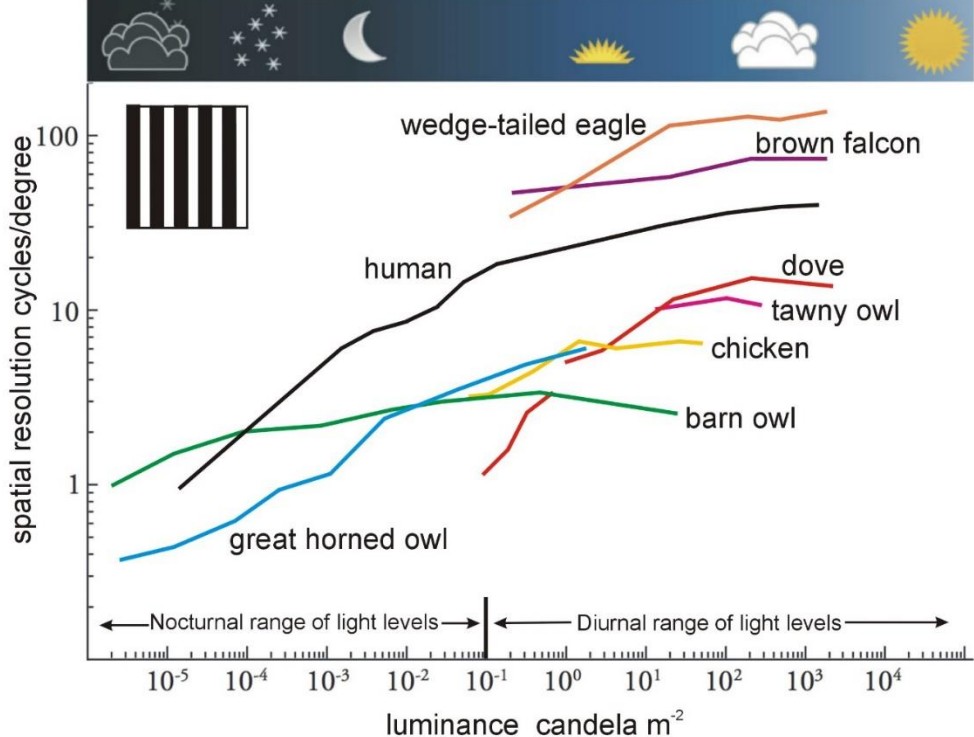

**Figure 1.** Resolution and light levels. The effect of light levels on visual resolution is significant. Acuity is often measured or estimated at high daytime light levels, but in a few bird species acuity has been measured across almost the full range of naturally occurring light levels. In some species, acuity has been determined across the narrower range of the light levels that occur in daytime and in some species across the daylight-twilight range. This figure shows some key points about the effect of light levels on spatial resolution. In all species shown here acuity has been determined using behavioural training techniques. Although the maximum acuities of these different species

are significantly different (for example, the maximum acuity of the eagle is 60 times higher than that of the dove) it is also clear that in all species acuity decreases considerably as natural light levels fall. The only species in which there is not a steep decline in acuity are Western Barn Owls *Tyto alba*, but even in Barn Owls acuity shows a significant fall as light levels decrease to the lower ranges of night-time. In some species, the decline is particularly steep and may be a significant reason why some daytime birds, such as doves, go to roost as light levels fall. (From [1], Figure 2.16).

A good rule of thumb is that visual resolution during twilight can be reduced by up to 10-fold compared with maximum resolution (acuity) [2]. This means that an animal would have to be 10 times closer to detect a given object at twilight than at midday. While this is confirmed by our everyday experiences, it applies equally to birds and will be behaviorally significant to birds which readily fly during twilight [16].

This reduction in resolution with light level applies to all diurnally active birds (Figure 1), and it has been argued [1] that many birds go to roost at dusk primarily because their visual resolution drops sufficiently that they cannot negotiate obstacles safely. (Note that owls have a different set of sensory and behavioural strategies that allow them to be nocturnally active within complex, woodland-type, habitats. However, their spatial resolution is low (Table 1 and Figure 1) and they may rely on only partial visual information when active at nighttime light levels [1].

### 2.5. Fields of View

The human field of view is unusual. Few other animal species exhibit the same features. Two eyes look straight out from the face. This results in a view of the environment that at any one instant takes in little more than half of the world about the head. The world always appears to lie in front of us and very little can be detected above and below, even in the frontal field. Moreover, the direction in which spatial resolution is highest lies directly ahead and we see less detail in our visual periphery [1].

This is quite unlike the field of view of birds. In all bird species the eyes are placed laterally in the skull (Figure 2) and together view a large sector of the world about the head (Figure 3). Some species, including some ducks, achieve total panoramic vision. That is, they see 360 degrees around the head and the full hemisphere above it. On the other hand, some of the larger birds of prey (Accipitridae) and bustards (Otididae) have been shown to have such extensive blind areas above their heads that when they pitch their head forwards to scan the ground below, they are flying blind in their direction of travel (Figure 3) [17]. Furthermore, in birds, the direction of highest spatial resolution projects approximately along the optic axis of each eye (Figure 2) with the result that birds have two directions of highest acuity, one on either side of the head, not directly in front of it.

The visual significance of these arrangements is that in the majority of birds visual information can be gained at any instant from an extensive volume about their head and that there are at least two regions of enhanced resolution in their visual field. However, these regions project laterally rather than directly forwards. The result of this is that to examine an object with the highest acuity, a bird is required to turn its head and appears to look at the object "sideways" rather than examine it with the head facing forwards. The main differences between the human and bird view of the world can be summed up as: humans have the impression that they move forward "into" the world and are usually moving towards a point where their spatial resolution is highest. Birds, on the other hand, can be considered to be flowing "through" the world. For the majority of birds, objects appear in front, flow past, and disappear behind, and the region of highest spatial resolution is lateral to the head rather than directly in front.

The crucial role of binocular vision in birds is not in the detailed examination of something that lies ahead [8,18]. Objects of interest tend to be detected laterally using the region of highest resolution, and visual control is subsequently passed to the binocular field [19]. The binocular field functions primarily in the control of the direction of travel towards an object, and in determining the time to contact that object, rather than in determining the distance to that object. Both the direction of travel and time to contact are determined

directly from the pattern of the optical flow of images across the retina in each eye [20,21], and it is in the binocular region of each eye that the flow pattern expands symmetrically about the direction of travel [21–23].

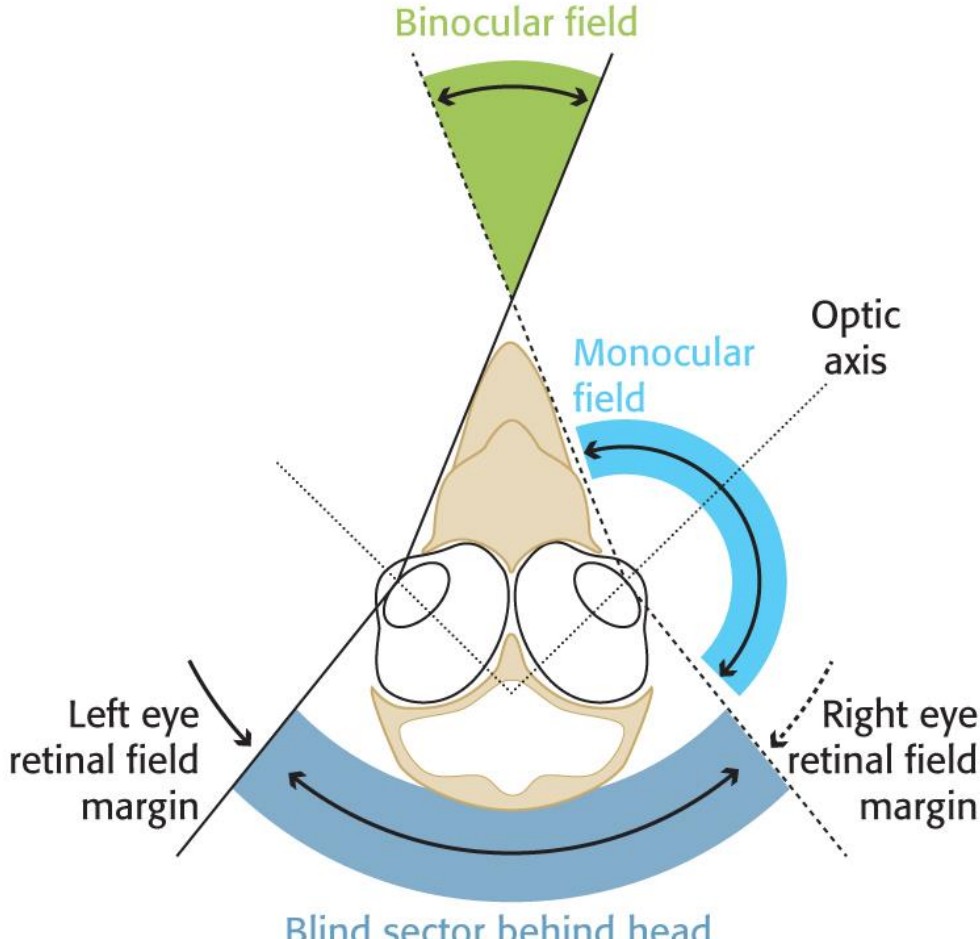

**Figure 2.** A diagrammatic section through the head of a bird showing a typical arrangement of eyes in the skull and how the visual fields of each eye combine. In all birds the eyes project laterally so that the axes of the eyes always diverge, no birds have forward-facing eyes. The field of view of each eye is combined to give the total field of view of the bird and to give a sector in front of the head where the fields of the two eyes overlap to give a binocular field. A wide degree of variation in these basic arrangements is found in birds and gives different degrees of binocular overlap and different width blind areas behind and above the head. Just small variations in the width of the field of view of each eye, and of eye position in the skull, can result in large differences in visual fields between species. The optic axis of each eye defines the approximate direction of highest visual acuity. Hence, birds have at least two areas of high spatial resolution that project into their lateral fields of view. (Diagram by the author and drawn by Nigel Hawtin (nigelhawtin.com, accessed on 5 December 2022)).

The region of binocular vision in birds is relatively narrow (typically between 20 and 40 degrees wide) and vertically long (100 to 180 degrees) [1]. In Canada Geese, the binocular field that lies directly ahead of a bird in flight is about 22 degrees wide [24]. In some ducks, the binocular field is only 10 degrees wide, but in some birds of prey and passerine species, it is 40–50 degrees wide. See [1] for a comparative table of binocular field width in a range of bird species. In humans, the binocular field is 120 degrees wide (Figure 3).

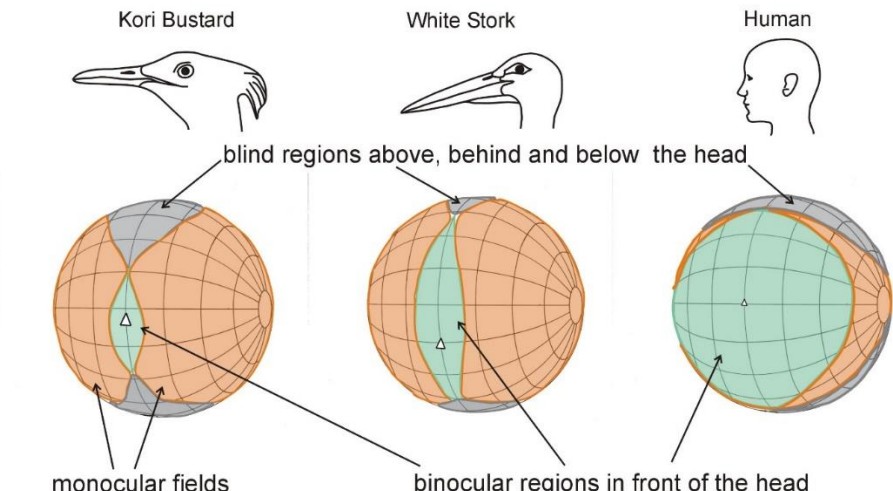

**Figure 3.** Depictions of the total visual fields of White Storks *Ciconia ciconia*, Kori Bustards *Ardeotis kori*, and Humans in which the key features of the field are shown as projected onto the surface of a sphere that surrounds the head. This presentation emphasizes how different the visual fields of these three species are. This is despite the visual fields of their individual eyes being similar. It is clear that differences in eye position in the skull, as well as differences in the visual fields of individual eyes, can result in quite different ways in which birds can extract information from the world about their heads. Kori Bustards are an example of a species which loses forward vision when the head is pitched forward to scan the ground below. White Storks and humans, on the other hand, maintain forward vision when the head is pitched forward through a large angle (Modified from [1], Figure 2.14).

## 3. Principles That Should Guide the Design and Deployment of Bird Diverters

Considering the characteristics of bird vision outlined above, it is possible to begin to describe the ideal characteristics of a power line diverter that will be effective for a range of bird species. It is also possible to consider how far apart diverters should be deployed along a power line.

### 3.1. Target Species for Power Line Diverters

Not all bird species are prone to collision with power lines and not all species can cause power outages if a collision occurs [25–28]. Faced with a diverter design and deployment problem what are the key factors to consider? Prime among these are: which species are the diverters aimed at, in which locations, at what time of day, and under what weather conditions? A key question is can there be an ideal, all-purpose, bird diverter? Can there be a diverter that is likely to be seen at a sufficient distance to allow any flying bird to detect it, interpret it correctly, and change its flight trajectory in sufficient time to avoid the hazard? Is there a good model species that diverters and their deployment can be focused around? If this model species can be diverted, then others should also be diverted. Is there a worst-case species and scenario situation?

A putative worst-case model species is Canada Goose *Branta canadensis*. They have the following attributes which make them prone to power line collisions. CanadaGeese are heavy (body mass, 4–5 kg). Like other heavy birds which use powered flight, they have a high cruising flight ground speed (17–19 m/s; 61–68 km/h) [29]. Along with this high flight speed, Canada Geese have low aerial manoeuvrability [5]. Furthermore, they will fly at dusk and dawn and in poor visibility conditions. They may fly regularly on commutes between roost sites and feeding sites, and they are highly social, rarely flying alone [16]. As a model species, Canada Geese do have the advantage that key aspects of their vision are known, which is not the case for the majority of bird species. They are known to be vulnerable to power line collisions [30] and are also a good surrogate for other wildfowl species also known to be vulnerable to collisions [26,28]

The visual resolution of Canada Geese is relatively low. Their visual acuity is 3.1 min of arc [13], i.e., the stripes of a black and white grating pattern have to be at least 3 min wide to be seen, narrower stripes cannot be resolved. This acuity is about 7 times lower than that of a young human with keen eyesight, and at least 3 times lower than that of mature adults or a person that wears correction spectacles for distance vision. However, the acuity of Canada Geese sits within the range of visual acuities known in birds (Table 1) (see [1] Appendix, 1 for a table of acuity determined in 46 species of birds). This difference between the acuity of a Canada Goose and that of a mature person would mean that if a human is able to just detect a particular object at a distance of 100 m, a Canada Goose would not be able to detect that same object until it is about 30 m from it. It is also worth noting that Canada Goose's visual acuity is 15 times lower than that of an eagle (Table 1). This means that if the goose can just detect a diverter at 30 m, the eagle would be able to detect it at 450 m. These differences in detection distances for a specific target are large and significant. They clearly demonstrate the need to have a clear idea of which species is being considered when designing a diverter.

### 3.2. Vision-Based Design and Deployment Considerations

As argued above, Canada Geese can be considered a worst-case species upon which to base power line diverter design and deployment. Data on the spatial resolution of Canada Geese and on their visual field parameters, provide useful clues that can aid the design of a diverter which needs to be detectable under a range of light levels and visibility conditions. The design of the diverter itself needs to take into account spatial resolution (at low light levels) and typical flight speed. The spacing between diverters along a power line needs to take into account the width of the binocular field and typical flight speed.

The basic requirement is that the diverter should be detected by a goose flying at twilight at a sufficient distance to permit the bird to change course and avoid a collision. If the diverter design can satisfy these requirements, it should also effectively divert geese at higher light levels. Importantly, it should also act effectively as a diverter for a range of other bird species which have a similar or higher spatial resolution, and similar or lower flight speeds.

### 3.2.1. High Internal Contrast

The range of background conditions against which a flight obstacle appears under natural conditions is highly variable. This variation arises because of changes in the brightness and colour of clouds, skylight, vegetation, and landscape. Even at high ambient light levels, clouds may vary from white to extreme black, and red colours can occur in twilight. It is important therefore to render an obstacle detectable through its own internal contrast pattern, which will be present regardless of the background conditions, rather than trying to make the obstacle contrast with the background. This principle is embodied in the design of warning and indicating patterns used in human collision-vulnerable situations, especially where a warning is to be effective throughout the full range of natural light conditions. Under these requirements, a simple high-contrast pattern of black and white is commonly employed. This ensures that regardless of the background, the warning pattern is likely to be conspicuous.

It is not advisable to use colour to generate the contrast pattern. While colour may appear attractive or salient to human observers, colour is detectable only at higher (daytime) light levels [3]. In addition, spatial resolution to chromatic patterns is always lower than to achromatic patterns [31,32]. This means that even under high light levels, a chromatic pattern will become detectable at a closer distance than the same pattern rendered achromatically. At low light levels (twilight and lower) a chromatic pattern is perceived in shades of grey. Therefore, although a chromatic pattern may have high detectability at daytime light levels, detectability is much reduced as light levels fall. An achromatic pattern will have higher detectability throughout the full range of naturally occurring light levels. This applies to the vision of all vertebrates and so it is always advisable to use achromatic

patterns to enhance detectability, especially if there is concern that collision-prone birds are active at lower light levels.

### 3.2.2. A Degree of Movement or Flicker

The salience of high-contrast flickering patterns is readily appreciated in everyday terms in humans when a waving flag is more readily detected than a static one. There is good evidence from physiological investigations of vertebrate vision that flickering single lights, high contrast oscillating patterns of gratings or checkerboards, and high contrast patterns with rotating elements produce high-amplitude physiological responses, measured directly from the eye using electroretinograms [33,34]. This has been demonstrated in the eyes of different bird species, most notably Rock Doves *Columba livia* and American Kestrels *Falco sparverius*, using rotating high contrast (black and white) patterns [35].

### 3.2.3. The Distance at Which a Diverter Pattern Is Detectable Should Be Guided by the Time at Which the Diverter Can Be Detected with Respect to the Approach of the Bird towards the Target

This is to allow the bird to not only detect but also process the information and then change its flight path to avoid the object. Clearly, the greater time before possible impact at which the diverter can be detected, the greater the chance of the bird changing flight direction successfully to avoid the obstacle. It would seem reasonable that a bird would require at least 2 s to take avoiding action/change flight path, but there is no field-based or experimental data to support this. If 2 s is used, then a diverter would have to be visible to a Canada Goose flying at its typical powered flight ground speed at least 36 m away from the diverter (Canada Goose flight ground speed, 17−19 m/s [29]). Thus, 36 m can be considered the minimum detection distance.

### 3.2.4. Assume That the Spatial Resolution Used to Detect the Diverter Is at Least 5 Times Lower Than Measured Acuity

This takes into account the fact that only rarely will the light levels at which birds fly be equal to those of bright daylight (the light levels to which threshold visual acuity refers; Table 1). Spatial resolution will decline significantly with light level (Figure 1) and a 5-fold decrease from maximum spatial resolution is highly likely for many birds in natural light conditions, especially close to twilight. Since the acuity of Canada Geese is 3 min of arc (Table 1), then a spatial resolution of 15 min should guide the size of the contrast pattern on the diverter. This means that the elements that provide the internal contrast of the diverter need to be 157 mm wide to be detected at a distance of 36 m. This is calculated using simple geometry, i.e., the width of a pattern element which subtends 15 min of arc at the minimum detection distance of 36 m.

### 3.2.5. Assume That Two Adjacent Diverters Should Be Just Visible in the Bird's Binocular Field at the Minimum Detection Distance

This requirement should ensure that in any view ahead at least one diverter will fall within the binocular field. Wider spacing could mean that a diverter could lie directly ahead of a flying bird but not be detected in the binocular field. This requires that the diverter is seen in the binocular field and addresses the proposition that it is the image falling within the binocular field that birds use to directly determine their direction of travel towards, and time to contact, an object that lies ahead [8]. In the case of a Canada Goose, this requirement means that diverters need to be placed every 12.5 m along a line. This is also calculated using simple geometry; the binocular field is taken to be 20 degrees wide in the direction of travel, while the minimum distance at which the diverters should be detected is 36 m.

## 4. An Ideal Power Line Diverter?

While it is possible to determine the width of the elements that make up the internal contrast pattern within a diverter (157 mm) there remains the problem of how many

elements are necessary for the pattern to be conspicuous. Pattern elements need to be repeated so that there are several contrast edges within the diverter. The number of contrast edges is maximized in a checkerboard design. Clearly, the larger the number of repeated elements, the greater probability that it will be detected. However, there must be a compromise with respect to the physical size of a diverter that can be attached to a power line in terms of both weight and wind resistance.

A possible minimum specification is that a diverter should incorporate a 3 × 3 checkerboard pattern. This would provide 12 contrast edges. Thus, for the ideal diverter in which each square of the pattern is 157 mm side, the overall pattern would need to be a 471 mm square. Combining this internal contrast requirement with the need for a degree of flicker in the pattern can be achieved with either oscillation or rotation of the device. Such movements would be best driven by wind rather than a motor. All of these design requirements can be combined in devices of the kind depicted schematically in (Figure 4A–D). One possible engineered solution is suggested in Figure 4E.

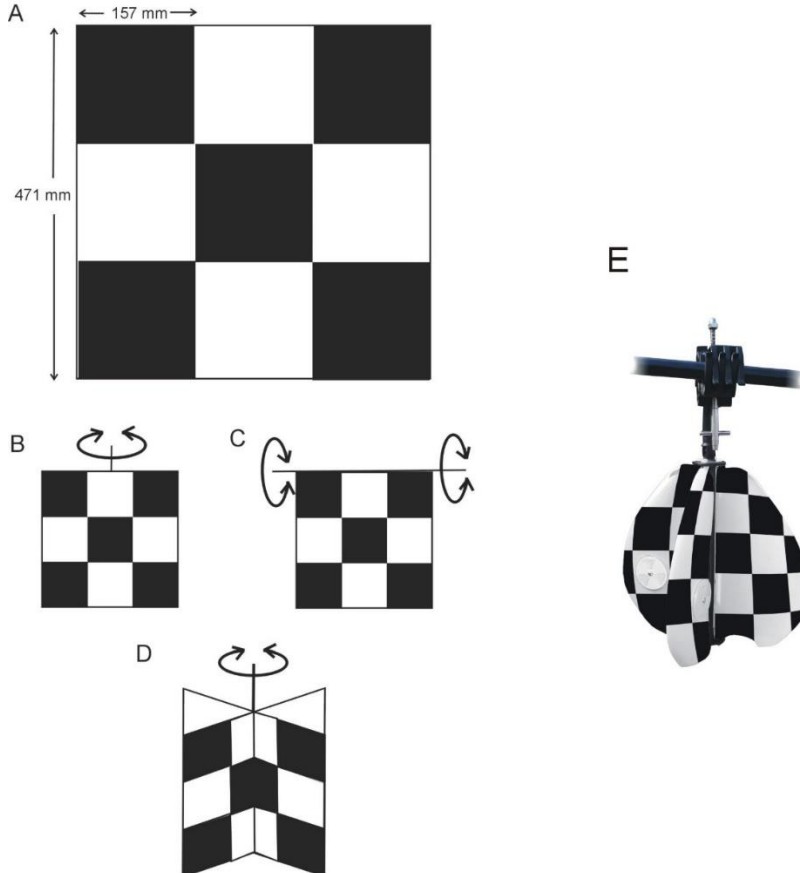

**Figure 4.** (**A**), essential design elements of a worst-case bird diverter based upon the vision of a Canada Goose. (**B–D**), depicts possible ways that the checkerboard pattern could be made to oscillate or rotate. The dimensions given in A indicate the physical size that should ensure a flying Canada Goose would be able to detect the diverter at a sufficient distance to change its flight trajectory and avoid the obstacle. (**E**), a possible interpretation of how the design features could be combined in a device.

The contrast of the achromatic patterns can be enhanced by careful choice of surfaces with respect to their reflection and absorption within the spectrum. About half of all extant bird species (Passerines Passeriformes, Parrots Psittaciformes, Gulls and terns Laridae) possess vision that extends into the UV portion of the electromagnetic spectrum; their spectral sensitivity extends from approximately 300 to 700 nm [1]. All other bird species have a spectral range of approximately 400–700 nm (similar to the human visible spectrum). To

gain maximum contrast in achromatic patterns, black sections should be highly absorbing, and white should be highly reflective, across the full spectrum of incident light. Given that some collision-vulnerable species have spectral sensitivity that extends from 300 to 700 nm, it is advisable to ensure that an achromatic pattern absorbs and reflects across this full spectral range. This will ensure that contrast is always maximised regardless of the spectral distribution of the ambient light.

*Canada Goose an Ideal Worst-Case?*

Canada Geese were chosen for this discussion of power line diverter design for several reasons. They are heavy and known to cause damage when they interact with power lines [36]. They are a good surrogate for other wildfowl, especially other geese, swans and larger ducks, which are also prone to power line impacts, especially because of their propensity to fly at low light levels. All these birds have low spatial resolution, high flight ground speed and low manoeuvrability. Basically, if a goose can be diverted by such a device, then many, possibly most, other bird species should also be diverted. Diverters that embody the same basic requirements (high internal contrast and movement/flicker) could be deployed at different sizes and spacings specifically to address problems with other species based on their known visual resolution, flight speeds, and binocular field widths. For example, target birds which have higher spatial resolution would be able to detect diverters with patterns containing smaller elements or at greater distances than the geese, or birds with broader binocular fields should detect and avoid diverters deployed less frequently along a power line. However, rarely will the flight and vision characteristics of other species be known well enough to design specific diverters. It would always seem safe to assume at least a putative worst-case similar to Canada Geese that has been employed here. A further caveat is those birds which can lose vision in the direction of travel by pitching their head forward to examine the terrain below, such as raptors [17] and bustards [37] (Figure 3). These birds are known to be prone to collisions with power lines and other human structures [38–40] and this is despite raptors having visual resolution considerably higher than that of Canada Geese (Table 1). To be salient for such species, diverters which are physically larger and more frequently deployed may be necessary to increase the probability that the hazard is detected in those intervals when the flying birds are looking forwards.

**5. Conclusions**

It has been argued here that the design and deployment of power line bird diverters should be based on the vision of birds rather than humans. While bird species differ in all aspects of their visual capacities, general principles that describe the vision of birds are now sufficiently understood to suggest some generic design and deployment rules. These take account of the spatial resolution of birds, how it is influenced by light levels, and how the binocular field is used to control the direction of travel and time to contact a target. Field trials of diverters based on these design considerations are now required.

**Funding:** This research received no external funding.

**Institutional Review Board Statement:** Not applicable.

**Data Availability Statement:** Data is contained within the article.

**Conflicts of Interest:** The authors declare no conflict of interest.

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
