# Peer review of "Vision-Based Design and Deployment Criteria for Power Line Bird Diverters"

_2673-6004, doi:10.3390/birds3040028_

Round 1

Reviewer 1 Report

The paper reviews extensively the different aspects in the vision of birds with respect detecting power lines. Based on the birds sense of vision the author proposes the minimum requirements for a device to prevent birds from power line collisions. The paper is very educational for the readers.

The very lengthy introduction into the different aspects of vision related to (power line) collisions is a nice review about vision of birds, but lacks the discussion of current diverters in the introduction. Both current as well as proposed diverters are not tested nor discussed (although different possible caveats are given for the proposed system). The only reference related to current diverters mentioned in the paper (1) seems to be a report which is not available on the internet, while there are several papers on the efficacy of diverters to reduce bird mortality. The way the paper is written now, looks more like a well-designed project proposal. One might expect if even a system is designed/proposed the author would also show the (first) results. All of this is lacking and that’s a pitty.

The article of “Hausberger 2018 Wide-eyed glare scares raptors: From laboratory evidence to applied management” might be of interest to the author. This paper describes the development and experimental testing of a super stimulus for birds based on visual aspects to deter birds of prey, corvids and other birds from airports.

Detailed comments:

Ln 79: sound and vision I would suggest; bird deterring on airports is based on both sound and vision queues; sound does scare away birds, e.g. distress calls

Ln 88: described -> describe

Ln 148: reference is missing

Ln 158: a reference which shows that Canada goose are prone to power line collisions would help too. The selection criteria given in the following lines hold for many more species (maybe even all waterfowl).

Ln 200: source of figure 2?

Chapter 10 (ln 379). According to the author one solution for a system is engineered and presented in figure 5. That’s very beautiful, but here the paper should actually start with results and findings.

Reviewer 2 Report

The paper is well written, and makes a novel case for specific designs based on the vision limitations of birds. The parallels drawn in no. 4 (page 3&4) between human driving behaviour and bird flight, is a good analogy. I would argue that large bustards fit the description of "worst case" species better than geese, given their limited vision, large size, fast powered flying speed, and - importantly - because no one has been able to come up with a very effective bird diverter for bustards. I understand that not enough may be known about bustard vision to fulfil the requirements of this study, however I would like to see this exercise repeated using bustards as a worst-case model. Our own studies have revealed that bustards do sometimes fly at night, and collision mortality data show that they are (at least in our region), the birds most heavily impacted by power line collisions.

One aspect that has not been addressed by the paper is the effect of weather conditions. How would the modelled parameters of the ideal bird diverter be affected by, for example, the presence of heavy fog or mist. Given that the goose is a waterbird that regularly commutes between waterbodies, such conditions may be likely. Would the black and white checkered bird diverter be sufficient in such situations, or would a diverter equipped with flickering lights further enhance its effectiveness?

Some comments suggestions are given below:

Line 18: "The recommended spacing of diverters along a power line are based...". Suggestion: "...is based..."

Line 30: Perhaps it would help the reader to define what is meant by "absolute and relative ability"

Line 35: (Figure 1) Maybe add an example of a device that is equipped with LED lights, e.g. the 'Overhead Warning Light' (OWL) from Preformed Line Products (RAPTOR CLAMP™ / OWL | Preformed Line Products)- these are used extensively in southern Africa.

Line 42: Is this statement true for the Rotamarka (H in Figure 1), in which I understand the author may have had some input?

Line 45: "...and there was insufficient known about bird vision...". This does not read well. Suggestion: "...and there was insufficient knowledge about bird vision..."

Line 46: Same as above: "...that sufficient is now known about bird vision...". Suggestion: "...that there is now sufficient knowledge about bird vision..."

Line 51: "...there would now seem to be...". Suggestion: "... there now seems to be..."

Line 74: "...and by power distribution network operators." The word distribution, in the context of power utilities, refers to only one aspect of the network. The distribution network generally refers to low to medium voltage power lines, while the transmission network refers to high voltage power lines. Many birds also collide with transmission line cables, and so it is important to note that the problem is one not just affecting the distribution network. Suggestion: "...and by power utilities."

Line 134: "...to warm us...". Suggestion: "...to warn us..."

Line 152: "A key question is, can...". Suggestion: "A key question is: Can..."

Line 181: Table 1. Do you have visual acuity information for bustards? If so, I would like to see this included in the table.

Line 254: Figure 4. I believe that at least some parts of this diagram appear in a previous paper by the author. Perhaps it is good to cite that paper in the caption?

Line 259: "The Bustard..." Which bustard? If reference is made to all bustards here, the 'b' should not be capitalized... If reference is made to the Kori Bustard depicted in the diagram, then the full name should be given.

Line 302: Why twilight and not night time? Waterfowl sometimes fly in the dead of night...

Line 326: Do you suggest that bird diverters incorporating LED lights should use only white light instead of colours? If so, then this is an important consideration and should be mentioned somewhere, e.g. in the discussion.

Line 339: There should be a period before the word "This".

Line 346: Add a comma after the word "detected".

Line 353: Yes, it is important to emphasize the minimum detection distance, and that a diverter should always strive to draw a birds attention well before this distance is reached.

Line 355: Why five times lower and not 10 times lower. If a worst-case scenario is assumed, then why not 10 times as mentioned on page 5?

Line 375: Does the Canada Goose have the worst binocular visual overlap measured in birds? If not, then I would suggest that this may not be the worst-case species on which to model bird diverter spacing...

Line 384: What will be the effect if even more checkered patterns are added? Is there an ideal size of checkered squares?I would image that there may be a point where the checkered blocks are so small that, when the diverter spins in a very high wind velocity, the effect of flickering in checkered pattern may eventually 'merge' into grey...

Line 419: Again, perhaps the model should look at even lower light levels than what is available at twilight, as many waterfowl sometimes fly in the worst conditions (the middle of the nights).

Line 435: Perhaps an example should be given for bustards - what spacing is required to alert a bustard in flight of the presence of a power line? 

Round 2

Reviewer 1 Report

Reshaping the paper and remove some discussion points realy improves the paper. Now the paper reads like a very nice framework for new bird diverter design, which hopefully will find it's first pilot!

A few double spaces and an extra line before para 2.2 and the paper is ready to be publised!

Author Response

Reply: Thanks for your support. I could not find the double spaces,  but extra line inserted before para. 2.2

Reviewer 2 Report

I have read the responses to my review, accept them, and am happy for the paper to be published in its current form.

Author Response

Thanks for your support. No changes necessary.